# Broadband sound barriers with bianisotropic metasurfaces

Bogdan-Ioan Popa [1], Yuxin Zhai [1] & Hyung-Suk Kwon [1]

Noise is a long standing societal problem that has recently been linked to serious health consequences. Despite decades of research on noise mitigation techniques, existing methods have significant limitations including inability to silence broadband noise and shield large volumes. Here we show theoretically and experimentally that acoustic bianisotropic materials with non-zero strain to momentum coupling are remarkably effective sound barriers. They surpass state-of-the-art sound isolators in terms of attenuation, bandwidth, and shielded volume. We implement our barriers with very compact active meta-atoms that owe their small size to their local response to external sound. Moreover, our active approach is not constrained by feedback stabilization requirements, in stark contrast with all traditional active sound control systems. Consequently, bianisotropic sound barriers have the potential to revolutionize noise control technologies and provide much needed solutions to an increasingly important and difficult challenge.

---

[1] Department of Mechanical Engineering, University of Michigan, Ann Arbor, MI 48109, USA. Correspondence and requests for materials should be addressed to B.-I.P. (email: bipopa@umich.edu)

The accelerated industrialization of the last several centuries has elevated noise from ubiquitous annoyance to health hazard correlated, among others, to an increased incidence of cardiovascular diseases[1]. Reducing noise has proven a difficult challenge because noise tends to be broadband and omnidirectional. Numerous passive solutions have been tried. Broadband composite materials such as sound absorbing panels present in many buildings are bulky and their effectiveness is severely limited especially at low frequencies[2,3]. Resonant structures are effective in reducing noise but have very narrow bandwidths[4–6]. Sound diffusion is used to alleviate noise effects by redistributing directional sound omnidirectionally, and not by absorbing the overall sound[7,8]. To overcome the limitations of passive sound absorbers, active solutions have been proposed[3,9]. The computational complexity involved in stabilizing the feedback loops between various active components makes these solutions effective only in limited volumes and cover relatively narrow frequency bandwidths on the order of 1–2%. For instance, noise cancellation headphones[10,11] are good at reducing the noise produced by jet engines but appear to amplify the conversations of fellow passengers.

Advances in material science and manufacturing provide a fresh approach toward achieving sound and vibration isolation. The mechanical behavior of matter has been described for a long time in terms of constitutive relationships that express stress versus strain and momentum versus particle velocity. This traditional view of materials has been challenged several decades ago when it was shown that, in general, stress can be caused by particle velocity and momentum may be generated by strain[12–22]. Materials satisfying this generalized coupling have initially been called Willis materials[12] and later bianisotropic materials by analogy with the bianisotropic magneto-electric coupling in electromagnetics[17]. These unusual bianisotropic (Willis) media have been shown to support rich dynamical behaviors. For instance, transformation elastodynamics revealed that bianisotropy could enable extraordinary control over the propagation of mechanical waves including the ability to cloak regions of space from detection with elastic waves[13]. Moreover, the first experimental acoustic bianisotropic materials[15,18,21] demonstrated ability to independently control the transmission and reflection characteristics. However, the properties of mechanical bianisotropy explored so far only touch the surface of what is possible and many properties are still to be discovered.

Here we demonstrate theoretically and experimentally the ability of a particular flavor of bianisotropic (Willis) materials to act as effective broadband sound barriers. We depart from the passive bianisotropy analyzed so far[12–22] and consider media in which only the strain (monopole) to momentum (dipole) Willis coupling term is non-zero. For the first time, we achieve this unusual type of coupling in active meta-atoms that feature local acoustic responses to impinging waves. This is in contrast with all previously reported bianisotropic metamaterials[15,18,21] that are based on non-local responses requiring large unit cells. The local nature of the acoustic response enables very compact broadband meta-atoms unachievable in passive Willis materials. Therefore, the bianisotropic metamaterials demonstrated here are excellent for low-frequency sound control in very compact devices. Finally, we show that bianisotropy leads to active meta-atoms composed of sensor-driver transducer pairs that require no feedback stability mitigation, which is unlike all active sound and vibration control systems demonstrated so far. The bianisotropic sound barrier concept is illustrated experimentally to demonstrate its potential.

## Results

**Active bianisotropy for sound isolation**. We focus here on the scenario of general interest outlined in Fig. 1a, in which a section of a vertical wall is replaced by a bianisotropic metasurface designed to prevent sound from propagating through the wall. Our goal is to compare side by side the effectiveness of the conventional wall and metasurface as sound isolators and to show that the latter performs significantly better and leaves a significantly quieter region behind it.

To understand the potential of acoustic bianisotropy to produce efficient sound isolators, we take a closer look at the behavior of bianisotropic fluids subjected to an incident external sound wave characterized by pressure $p_i$ and particle velocity $\mathbf{v}_i$. Recent metamaterial research has shown that a microscopic description of this behavior based on the polar response of the metamaterial unit cells (meta-atoms) is better suited for metastructures whose lattice size is roughly a tenth of the operating wavelength or larger[17,20,22–25] because it more accurately takes into account important physics such as spatial dispersion and edge effects. According to this description, the meta-atom is modeled as a collection of monopole, dipole, and coupled monopole-to-dipole and dipole-to-monopole sources (inclusions) placed in a homogeneous background of characteristic impedance $Z_0$. These sources generate net pressure or particle velocity in response to the local pressure $p_{loc}$ and local particle velocity $\mathbf{v}_{loc}$ according to $p_m = \alpha_m p_{loc}$, $\mathbf{v}_d = \overline{\overline{\alpha}}_d \mathbf{v}_{loc}$, $\mathbf{v}_{md} = \boldsymbol{\alpha}_{md} Z_0^{-1} p_{loc}$, $p_{dm} = Z_0 \boldsymbol{\alpha}_{dm} \cdot \mathbf{v}_{loc}$, where the "m" and "d" indexes refer to the conventional monopole-to-monopole and dipole-to-dipole sources, the "md" and "dm" indexes refer to the monopole-to-dipole and dipole-to-monopole bianisotropic coupling. The scalar $\alpha_m$, vectors $\boldsymbol{\alpha}_{md}$ and $\boldsymbol{\alpha}_{dm}$, and second order tensor $\overline{\overline{\alpha}}_d$ are the polarizabilities that quantify the linear dependency between the fields generated by the meta-atom sources and the local fields $p_{loc}$ and $\mathbf{v}_{loc}$.

Essentially all past studies on mechanical bianisotropy considered reciprocal passive materials in which the monopole-to-dipole ($\boldsymbol{\alpha}_{md}$) and dipole-to-monopole ($\boldsymbol{\alpha}_{dm}$) couplings are correlated to enforce reciprocal wave dynamics[14,17]. Here we show that breaking this condition in an active metasurface in which $\boldsymbol{\alpha}_{md} \neq 0$ and $\boldsymbol{\alpha}_{dm} = 0$ leads to very efficient sound barriers. The meta-atom satisfying the above constraints is shown in Fig. 1b, which highlights the three types of sources involved in this flavor of bianisotropy. We show in Supplementary Note 1 and Supplementary Fig. 1 that the transmitted field through a bianisotropic metasurface in which the monopole-to-dipole source is polarized perpendicular to the metasurface ($\boldsymbol{\alpha}_{md} = \alpha_{md}\hat{\mathbf{x}}$, where $\hat{\mathbf{x}}$ is the unit vector normal to the metasurface) and the conventional dipole response is isotropic ($\overline{\overline{\alpha}}_d = \alpha_d \overline{\overline{\mathbf{I}}}$, where $\overline{\overline{\mathbf{I}}}$ is the second order unit tensor) can be written as

$$p_t = \frac{1 + \alpha_{md} - \alpha_m \alpha_d}{(1 - \alpha_m)(1 - \alpha_d)} p_i. \qquad (1)$$

Equation (1) shows why bianisotropic media are much better for sound isolation than conventional materials. In conventional materials $\alpha_{md} = 0$ and the transmitted field disappears when $\alpha_d \alpha_m = 1$. This only happens for infinitely stiff and dense walls for which $|\alpha_d| = |\alpha_m| = 1$ or in gain media in which $|\alpha_d| > 1$ or $|\alpha_m| > 1$. The former case is not physical and the latter corresponds to situations in which the meta-atom sources produce fields more intense than the locally sensed fields. Traditional active sound and vibration control methods have been employed to produce these large active responses, but the previously reported methods proved prone to instability and unsuitable for broadband sound and vibration isolation[3].

Bianisotropy solves this limitation. According to Eq. (1), $p_t$ becomes zero when $\alpha_{md} = \alpha_m \alpha_d - 1$. Given the high contrast between the density and elastic moduli of a conventional wall and air, the magnitudes $|\alpha_d|$ and $|\alpha_m|$ are close to unity, and only a

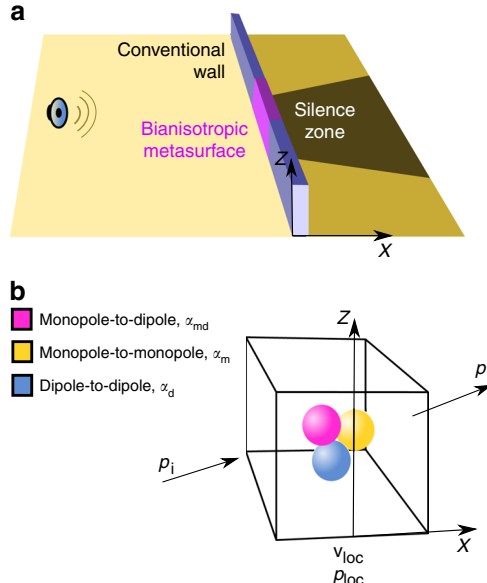

**Fig. 1** Bianisotropic metasurface sound barrier. **a** A section of a conventional wall is replaced by a bianisotropic metasurface window designed to prevent sound from propagating behind the metasurface. **b** The metasurface unit cell (meta-atom) is made of three types of scattering sources (inclusions) whose acoustic responses are described in terms of bianisotropic polarizability $\boldsymbol{\alpha}_{md}$ and conventional polarizabilities $\alpha_m$ and $\overline{\overline{\alpha}}_d$. The local pressure and particle velocity at the position of the inclusions are $p_{loc}$ and $\mathbf{v}_{loc}$

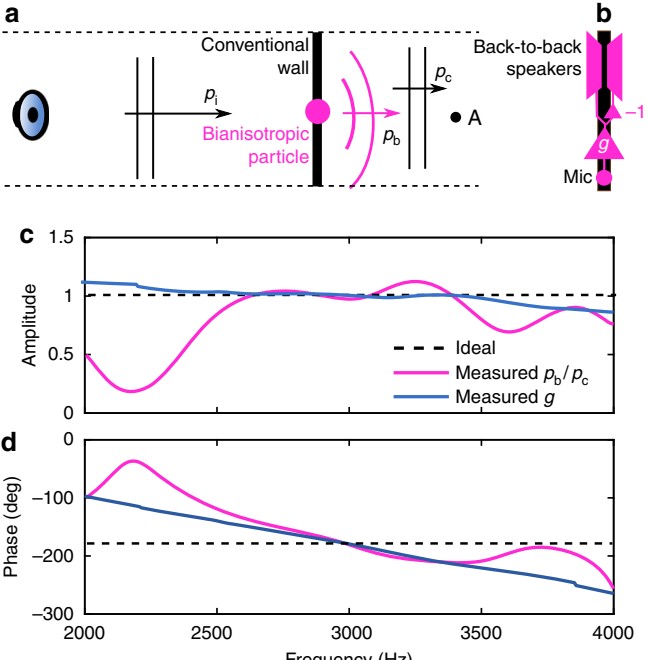

**Fig. 2** Bianisotropic meta-atom design. **a** A bianisotropic meta-atom is embedded in a conventional wall. An incident wave $p_i$ produced by an external speaker propagates through the conventional wall. The bianisotropic response of the particle, $p_b$, and the transmission though the wall, $p_c$ are recorded at point A situated 2 cm behind the particle. **b** The bianisotropic meta-atom is composed of monopole and dipole transducers connected through the amplifier of impulse response $g$. The dipole driver is implemented as two back-to-back speakers driven 180° out-of-phase and generates pressure fields of inverse polarity in the two semi-planes separated by the dipole's anti-symmetry plane. The monopole sensor is placed in the antisymmetry plane of the dipole source and senses zero net pressure generated by the dipole. **c**, Amplitude of $p_b/p_c$ measured on the meta-atom transmission side 1 cm behind the meta-atom versus measured amplitude of amplifier impulse response $g$. **d**, Measured phases of $p_b/p_c$ and $g$

small amount of monopole-to-dipole coupling ($|\alpha_{md}| \ll 1$) is needed to enforce $p_t = 0$. Physically, the pressure of the sound wave propagating through a conventional wall in the absence of bianisotropy is $p_c = (1 - \alpha_m \alpha_d)(1 - \alpha_m)^{-1}(1 - \alpha_d)^{-1} p_i$, which represents the transmitted pressure $p_t$ given by Eq. (1) when $\alpha_{md} = 0$. The bianisotropic sources embedded into the wall produce an additional transmitted acoustic wave whose pressure $p_b = \alpha_{md}(1 - \alpha_m)^{-1}(1 - \alpha_d)^{-1} p_i$ can be tuned via the choice of $\alpha_{md}$ to remove the conventional transmitted sound $p_c$. We note that $p_t = p_b + p_c$ as shown in the Supplementary Note 1. Although $\alpha_{md}$ needs to be generated actively because $\alpha_{dm} = 0$ breaks the reciprocity condition of passive materials[12–14,17], the generated bianisotropic response is much smaller than the locally sensed $p_{loc}$ because $|\alpha_{md}| \ll 1$. Therefore the material can be made broadband without compromising stability.

**Active bianisotropic meta-atom design**. The above analysis shows that we need to embed bianisotropic sources implementing the required $\alpha_{md}$ into a conventional wall to make it impervious to sound. However, these sources have to satisfy a very difficult to achieve set of constraints. Namely, each inclusion (a) must produce a velocity field in response to the sensed pressure, i.e., the inclusion implements the monopole-to-dipole coupling; (b) must produce a velocity field that does not contribute to the local pressure field $p_{loc}$ and thus it is not sensed back by the inclusion, i.e., the inclusion has no local feedback; (c) must have a local response. We realize these constraints, as follows.

Functionally, each meta-atom has a monopole sensing transducer that couples to the local pressure field (Fig. 2a). The produced electric signal is amplified by an electronic circuit having the impulse response $g$ defined as the ratio of the output-to-input voltages. The electronic circuit drives a dipole transducer implemented as two 180° out-of-phase speakers. The unidirectional nature of the amplifier connecting the sensing and driven

transducers enforces the requirements $\alpha_{md} \neq 0$ and $\alpha_{dm} = 0$. Moreover, the monopole transducer is placed in the antisymmetry plane of the dipole transducer to cancel the mechanical coupling between the two. Consequently, the velocity field produced by the dipole is not heard by the monopole, and the local feedback loop is removed (see Fig. 2b). The Supplementary Note 2 and Supplementary Fig. 2 provide the fabrication details of the meta-atom.

In this design the amplifier impulse response $g$ is proportional to the bianisotropic polarizability $\alpha_{md}$, which provides a simple way to tune $\alpha_{md}$ as follows. One bianisotropic meta-atom is embedded in a 6-mm-thick polycarbonate wall, as illustrated in Fig. 2b. The meta-atom is designed so that its conventional (non-bianisotropic) response matches that of the polycarbonate wall as closely as possible (i.e., the meta-atom $\alpha_m$ and $\alpha_d$ match those of the wall) to simulate our scenario in which bianisotropic polarizabilities are embedded into the wall. This requirement also minimizes the diffraction at the boundary between the meta-atom and polycarbonate wall. A short Gaussian pulse is sent through the wall and the pressure $p_t = p_b + p_c$ on the other side is measured 10 cm behind the bianisotropic meta-atom at point A marked in Fig. 2b. In addition, the pressure propagating through the bare polycarbonate wall is measured at point A to obtain $p_c$. From these measurements we compute $p_b/p_c = p_t/p_c - 1$. Our design goal is to make $p_t = 0$ and thus $p_b/p_c = -1$. Since $p_b \sim \alpha_{md} \sim g$ this is achieved by iteratively modifying the impulse response

$g$ to obtain $p_b/p_c = -1$ at 3000 Hz. We chose this frequency because it is in the middle of a band in which the sensing and driven transducers have relatively flat acoustic responses.

Figures 2c, d show the amplitude and phase of $p_b/p_c$ and $g$ measured in the 2000–4000 Hz octave. The plots confirm that $g$ is indeed proportional to the measured $p_b/p_c$ in the selected octave. The disagreements between the two sets of curves below 2500 Hz is caused by the non-flat acoustic response of the speakers used to implement the dipole transducer.

The two figures also show that the target $p_b/p_c = -1$ has been achieved at 3000 Hz, but the signal delay through the electronics determines a non-zero slope of the $p_b/p_c$ phase. As a result, the phase of $p_b/p_c$ departs from the ideal 180° away from 3000 Hz. Even though this limits the bandwidth to some degree, we show in the following that sound barriers based on the bianisotropic meta-atom have bandwidths at least one order of magnitude larger than current passive and active noise cancellation solutions of comparable isolation performance[3]. Moreover, we show that faster electronics significantly increase this bandwidth even further.

**Active bianisotropy versus active sound control.** The bianisotropic meta-atoms described above are unlike any other active sound control system reported to date. Most traditional active sound and vibration isolation solutions (see ref. [3] for an excellent review of the field) have significant performance limitations imposed by their need to actively control the feedback loops between the sensor and driven transducers. They typically employ several microphones and speakers spread out in a volume. These components are coordinated by a central computer that employs various signal processing algorithms to control in real-time the feedback loops occurring between all these elements. The complexity of these systems increase significantly with the number of microphones and speakers, thus they are not scalable in the number of speakers and microphones. Moreover, most active control systems are only designed for a given source of sound, e.g., for the narrowband sound produced by a jet engine. Consequently, traditional sound control systems are not suitable for reducing the diverse, complex, and ubiquitous noise engulfing human habitats. Moreover, the mechanical response of these systems are inherently non-local which makes their performance sensitive to the incoming excitation direction.

To avoid the limitations of centralized approaches, decentralized control methods that employ self-contained active unit cells have been proposed in the context of plate vibration control[26] and acoustic metamaterials[27–31]. However, the feedback between each unit cell's sensor and driver components is always present and reduce the effectiveness of these approaches. For instance, the feedback loop built into the unit cells used to reduce the vibration of plates[26] avoid injection of energy into the incident wave, which reduce the vibration control efficiency significantly.

In contrast, the bianisotropic meta-atoms do not suffer from the limitations imposed by feedback stability control and enable excellent broadband sound barriers. The increased bandwidth comes from the bianisotropic requirement that the velocity field produced by the meta-atom does not contribute to the local pressure field sensed by the meta-atom. This type of behavior is achieved here by placing the sensing transducer in the antisymmetry plane of the driven transducer and leads to compact meta-atoms in which the meta-atom's sensor and driver do not generate a local feedback loop. Moreover, non-local feedback loops may exist in scenarios in which the wave produced by the meta-atom is scattered back towards the sensing element. Remarkably, the stability of the bianisotropic meta-atom is maintained even in these scenarios without employing any supplementary feedback control mechanisms, as demonstrated in Supplementary Note 3 and Supplementary Fig. 3.

**Metasurface measurements.** Figure 3a shows the experimental setup that replicates the scenario illustrated in Fig. 1. A polycarbonate wall 6 mm thick is sectioned in the middle and a window 17.5 cm wide is replaced by the bianisotropic metasurface pictured in Fig. 3b and composed of five identical meta-atoms placed side by side. A speaker placed 40 cm in front of the wall acts as a quasi point source that excites short Gaussian sound pulses propagating through the wall. We measure the spatial-temporal distribution of sound behind the wall in the region highlighted in Fig. 3a using a standard raster scanning technique[32–35] to characterize the performace of the sound barrier.

Figure 3d, e shows the sound pressure level measured at 3000 Hz. When the barrier is activated (d) it produces a region of silence in which the sound amplitude drops significantly compared to the sound pressure level behind the conventional wall/deactivated barrier (e) in the entire region behind the barrier. To quantify sound suppression we compute in Fig. 3f the sound level difference measured with the barrier activated and deactivated. This quantity measures how much better is the metasurface at reducing sound penetration compared to the conventional wall.

Remarkably, we obtain sound suppression levels in excess of 15 dB in the entire region behind the barrier, which is significantly larger than the 1–3 dB obtained in broadband sound insulation materials of same thickness as the bianisotropic metasurface[2,3]. Moreover, the measurements demonstrate that the two requirements for sound isolation occur concurrently. First, the transmission coefficient through the barrier is low, i.e., $p_t/p_i \approx 0$. Second, the diffraction from the metasurface edges is small as demonstrated by the uniform field variation behind the metasurface. This is explained by the relatively small bianisotropic polarizability $\alpha_{md}$, which causes low intensity fringe fields generated at the metasurface edges. Since the unpowered metasurface is designed to match the acoustic properties of the conventional wall, the diffracted fields consist solely of the low intensity actively generated fringe fields. Due to low diffraction we expect the metasurface to be more effective at isolating sound than even an infinitely stiff and dense wall of the same size. Figure 3g illustrates this hypothetical situation. Even though no sound penetrates through an infinitely stiff wall (i.e., zero transmission coefficient), diffraction redirects significant energy behind the wall which drastically reduces sound isolation efficiency below that of the metasurface. Figure 3g shows the large diffracted fields engulfing the entire region behind the infinitely stiff wall.

The metasurface is designed for broadband operation and avoids resonant components. To quantify the operational bandwidth we define the sound suppression level (SSL) as the difference between the average sound pressure level (SPL) measured behind the barrier in the region labeled A (delimited by the lines pictured in Fig. 3f) and the average sound pressure level measured behind the conventional polycarbonate wall in the region labeled B. The white lines separating regions A and B were obtained by uniting the speaker position with the edges of the metasurface. Therefore, region A represents the area in which we expect to see an acoustic"shadow" cast by the metasurface. The figure of merit, SSL, has been chosen to quantify the acoustic shadow intensity by averaging the SPL difference in region A behind the metasurface and comparing it with the average SPL difference measured behind the conventional wall (i.e., measured in region B). Note that regions A and B do not change with frequency and are determined entirely by the geometry of the experimental setup. Figure 4a shows the measured SSL in the octave 2000–4000 Hz.

The metasurface reduces sound transmission more efficiently than an infinitely stiff wall of same length in a relatively large bandwidth of 20%. By comparison, previous passive and active

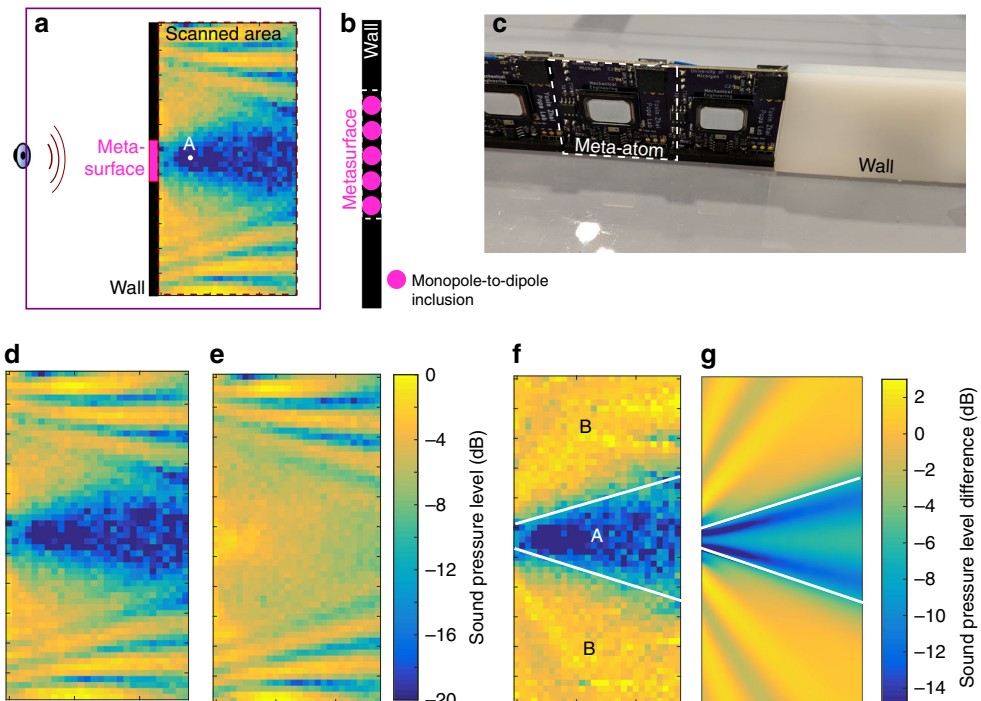

**Fig. 3** Experimental setup and performance. **a** A speaker placed on the edge of a two-dimensional acoustic waveguide sends acoustic waves inside the waveguide towards a polycarbonate wall 6 mm thick. The bianisotropic metasurface replaces a wall section 17.5 cm long in the middle of the wall. The sound pressure distribution is measured in a 110 cm by 50 cm region placed immediately behind the wall. The region shows the measured sound pressure level at 3000 Hz obtained with the activated metasurface. Point A is situated 10 cm behind the metasurface. **b** The metasurface is composed of five meta-atoms placed side by side. Each meta-atom simulates a monopole-to-dipole inclusion embedded into the polycarbonate wall. **c** Photo of metasurface and wall. The monopole-to-dipole meta-atom is highlighted. **d** Sound pressure level (SPL) measured at 3000 Hz with the bianisotropic response activated ($\alpha_{md} \neq 0$) and **e** deactivated ($\alpha_{md} = 0$). **f** SPL difference between the activated and deactivated states. The sound suppression level (SSL) is defined as the difference between the average SSLs computed in regions A and B. **g** Numerically simulated SPL difference for an infinitely stiff and dense wall

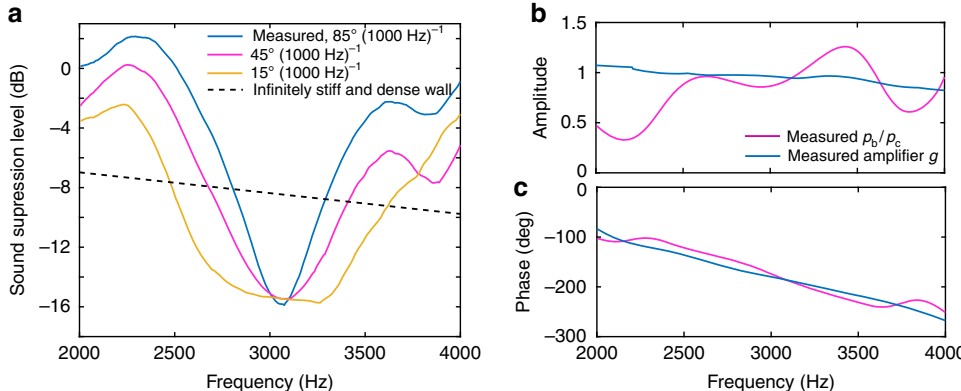

**Fig. 4** Measured sound suppression level. **a** Measured suppression level compared against the suppression levels expected for amplifiers having impulse response phase gradients of 45°(1000 Hz)$^{-1}$ and 15°(1000 Hz)$^{-1}$. **b** Amplitude and **c** phase of metasurface bianisotropic relative response $p_b/p_c$ measured at point A in Fig. 2a and compared against measured amplifier impulse response $g$

sound isolators of comparable performance had bandwidths an order of magnitude narrower, i.e., 1–2%[3,6]. To investigate how we can expand the bandwidth even further we compare the bianisotropic response of the metasurface $p_b$ against the conventional response $p_c$. Figure 4b, c illustrates the amplitude and phase of $p_b/p_c$ measured in the middle of the silence zone (point A in Fig. 3a) which confirms that the transmitted sound is almost zero, i.e., $p_t = p_b + p_c \approx 0$. More importantly, both the amplitude and phase of $p_b/p_c$ match very well those of the unit cell transfer function implemented in electronics ($g$ in Fig. 2b), which confirms that $p_b/p_c \sim \alpha_{md} \sim g$. The slight difference between $p_b/p_c$ and $g$ is

determined by the non-flat frequency response of the speakers and microphones embedded into the metasurface.

In addition, Fig. 2c, d obtained for one meta-atom match qualitatively and quantitatively the results obtained for the metasurface and shown in Fig. 4b, c, which confirms that the metasurface is scalable in the number of meta-atoms. Namely, identical meta-atoms can simply be stacked to increase the acoustically shielded area.

Figure 4c shows that $p_t \approx 0$ at 3000 Hz when the phase of $p_b/p_c$ is ~180°, but departs from the ideal value away from 3000 Hz. We can extend the bandwidth of the metasurface by reducing the

phase gradient using faster electronics. Figure 4a shows the expected SSL for various gradients between $15°(1000\ Hz)^{-1}$ and the measured $80°(1000\ Hz)^{-1}$, which confirms that faster electronics produce very broadband sound barriers. For example, a phase gradient of $15°(1000\ Hz)^{-1}$ produces better results than an infinitely stiff and dense wall in the entire band 2500 Hz to 3500 Hz. We also note that the metasurface increases the sound pressure level in the band 2000–2500 Hz by less than 2 dB. A vast literature on sound perception shows that 2 dB is a very small value situated around the smallest pressure difference detected by humans[36]. This small unwanted increase over the transmission through the bare polycarbonate wall is generated by the active bianisotropic response $p_b$ being in phase with the conventional response $p_c$. Since the bianisotropic polarizability $\alpha_{md}$ was chosen to insure that $p_c$ and $p_b$ have the same magnitude, it follows that the maximum sound produced by the metasurface is at most $2p_b$, which corresponds to a maximum sound pressure level increase of 3 dB. If needed, the increase can be canceled using faster electronics as predicted in Fig. 4a [see the $45°(1000\ Hz)^{-1}$ and $15°$ $(1000\ Hz)^{-1}$ curves] and/or by rejecting the gain band using one or two notch filters added to the meta-atoms' existing bandpass filters.

## Discussion

Active metamaterials implementing the sensor-driver architecture employed here have struggled for a long time with stabilizing the feedback loop between the driver and sensor[26,27,31,37] which led to designs that are narrow band and not scalable. The bianisotropic metasurface described here does not suffer from this issue. Due to the complementary nature of the monopole and dipole fields, the sensing monopole and driven dipole are naturally decoupled as long as their planes of symmetry are identical. Together with the requirement that the actively injected energy into the incident wave is much smaller than the incident energy ($|\alpha_{md}| \ll 1$), the sensor-driver decoupling translates into a metasurface that is unconditionally stable in a broad range of frequencies regardless of the topology of the environment.

The fabricated metasurface demonstrates experimentally that bianisotropic materials can be shaped into efficient sound barriers that prevent broadband sound from penetrating behind the barriers. The approach is general in that it applies to other noise mitigation devices such as anechoic terminations used to cancel reflections from walls and other boundaries. Another key benefit of our approach is that, unlike other active noise mitigation methods, the bianisotropic meta-atoms have a strictly local response to the local sound field and are thus self-contained and function independently from their neighboring meta-atoms. Therefore, the metasurface size and, consequently, shielded region can be easily extended by adding more meta-atoms to the metasurface.

Remarkably, we showed that bianisotropic barriers perform better than even infinitely stiff and dense walls in practical scenarios in which the barrier dimensions are constrained. Even though infinitely stiff and dense walls support zero transmission, diffraction from their edges can be significant. In contrast, the diffraction from the bianisotropic metasurface presented here is small and the transmission through it can be engineered to be negligible. We believe that the surprising benefits of mechanical bianisotropy discussed here provide a clear path forward toward solving long standing problems in noise and vibration control.

## Methods

**Experimental apparatus**. The experimental measurements are done inside a two-dimensional waveguide composed of two transparent polycarbonate plates 120 cm by 120 cm by 0.6 cm and separated by 3.5 cm. A polycarbonate wall 6 mm thick and 3.5 cm tall divides the interior of the waveguide into two regions. A window 17.5 cm is cut in this wall and replaced by the bianisotropic metasurface. The metasurface is composed of five meta-atoms and each meta-atom is built on a

3.5 cm by 3.5 cm printed circuit board (PCB). All the required electronics are mounted on the PCB except for the ARM Cortex M3 microprocessors. For increased flexibility each microprocessor is part of the commercial platform ArduinoDue and is connected to its meta-atom through three wires (input, output and ground). All unit cells are powered through 3 wires (±3 V and ground). The overall metasurface thickness including mounted components is ~9 mm. A sponge placed between the metasurface and the top waveguide plate minimizes the sound transmitted behind the wall, so that the acoustic impedance of the polycarbonate wall is on the same order of magnitude with the disabled (unpowered) metasurface, i.e., the metasurface and polycarbonate wall have approximately the same polarizabilities $\alpha_d$ and $\alpha_m$. A 3 cm tall commercial speaker is placed on the edge of the waveguide 40 cm in front of the metasurface and launches quasi-cylindrical waves toward it. The spatial and temporal distribution of sound is measured by a MEMS microphone that raster scans the area of the waveguide behind the wall/metasurface. The frequency spectrum of the transmitted sound is obtained from the time domain measurements via Fourier transforms.

**Simulation of infinitely stiff and dense sound barrier**. Comsol Multiphysics has been used to measure the sound transmitted through a conventional wall of same thickness and material properties as the wall used in the experiment (polycarbonate). The transmitted fields have been recorded behind the wall. We performed a second simulation in which a section of the polycarbonate wall has been replaced by a fictitious infinitely stiff and dense wall. The simulated fields obtained in the second simulation have been recorded and subtracted from the fields obtained with the polycarbonate wall by itself (first simulation). The SPL difference thus obtained is plotted in Fig. 3g.

## Data availability

The data that support the findings of this study are available from the authors on reasonable request, see author contributions for specific data sets.

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

## Acknowledgements

This work was supported by the University of Michigan through B.I.P.'s startup fund.

## Author contributions

B.I.P. has developed the concept and has written the manuscript. B.I.P. and Y.Z. designed the bianisotropic metasurface and analyzed the results. Y.Z. and H.K. fabricated the metasurface and performed the measurements.

## Additional information

**Competing interests:** The authors declare no competing interests.

