## [Peer Review File · Nature Communications]

Reviewers' Comments:

Reviewer #1:

Remarks to the Author:

This manuscript provides a numerical and experimental demonstration of the use of an active sound barrier that makes use of concepts associated with acoustic bianisotropy to demonstrate elevated levels of sound isolation when compared with idealized sound isolating partitions. The work builds upon current research on the topic of acoustic bianisotropy (also known as Willis media) and applies it to the specific case of sound isolation. The manuscript is well written and the results are of interest to a wide range of researchers in engineering and the physical sciences. While this is the case, there are several key points about the manuscript that must be addressed prior to acceptance and publication. What follows below is a list of primary and secondary points that should be addressed by the authors. The primary points are more broad-based questions on the scientific merit of the work while the secondary points are more focused on specific questions about the presentation of the content. If the concerns of the reviewer are properly addressed, and the claims presented in the present manuscript are borne out, the results presented here are of significant scientific interest and should be published.

The primary concerns with the submission are listed below.

1) The authors contend that it is solely the use of active bianisotropic elements that enable the significant improvements in sound isolations achieved in the results presented here. Indeed, equation (1) and the discussion surrounding it suggest that this is indeed that case. However, the authors have only shown the case of a finite wall with a finite window that is either of infinitely stiff and immobile "material" and therefore diffraction effects around the finite wall are present. In absence of diffraction, would this still be the case and why? The reason this question arises is that the formulation employed in Eqns. (4)-(7) imply that the wall itself acts as a scatterer of acoustic pressure. If the wall were infinite in extent and was infinitely stiff and immobile, then the impedance would be infinite and no pressure would be transmitted. This implies that the only way it is possible to improve upon the idealized finite length wall is to simply cancel the diffracted field. This seems to be the reason that the performance is somewhat narrow band and that the bandwidth of performance is dictated by the amplifier response of the active elements as indicated in Fig. 3. These concepts were not clearly fleshed out in the manuscript, though they should be since it goes to the core of the argument for the active bianisotropic metasurface. More detail needs to be provided in both the main article and supplementary material to address these points and place the behavior on more complete foundation.

2) Along the same lines, it is not clear to this reviewer that this necessarily represents an advancement over the traditional actively sound control either within a space or through apertures. Indeed, the argument that the "bianisotropic particle," which is simply a microphone with a dipolar source, is not entirely convincing. For example, if the sensor is placed on one side of a wall of finite impedance, and then the source is embedded in the same wall, then the source acts as a monopole in each half space, obviously one that has opposite polarization on each side of the wall. It is clearly a monopole (pressure) sensor and dipolar source. However, when this is embedded in a wall structure rather than acting in free space, as is the true concept of a α_{md} polarization, then it is not clear to this reviewer that it acts as a true monopole to dipole scatterer. The argument that this configuration can be used to mimic a bianisotropic particle needs to be improved and, further, it needs to be shown that what has been shown is not simply an implementation of a standard active noise control method.

a. This latter point is very important and also relates to the fact that the current manuscript does not address an existing literature on active noise control techniques or demonstrations. Note that it this reviewer does not believe that this manuscript would be un-publishable if the approach did have strong correlation with existing active noise control techniques since the current study investigates the potential for improving active noise control using ideas that geminated in metamaterial concepts. However, it is very important that the ultimate publication acknowledges this vast literature and is placed in the correct context in that domain.

b. In regards to the structure of the bianisotropic meta-atom, it is strongly suggested that a more detailed figure showing the structure of the barrier that was tested be provided. This is very important in order to justify the claims of bianisotropy and the use of the specific configurations provided in the paper.

The concerns raised below are more minor issues that should be addressed in any future publication. They are listed in the order they are encountered in the manuscript.

- 3) The authors often refer to sound "blocking," which seems to imply sound isolation (i.e. the elimination of transmitted sound without regard to what happens on the side of the incident sound), rather than absorption of sound. If this is the case, isolation is a preferred term and "blocking" should be eliminated from the manuscript.
- 4) There are minor errors in grammar and usage throughout the text. Please do a careful read of the document prior to re-submission.
- 5) The references provided in the introduction for passive sound isolation are all very new and primarily associated with metamaterials. In order to place this into the correct context, a few more seminal references on sound isolation in architectural acoustics should be provided to illustrate how long this has been a known problem.
- 6) In the first paragraph of the introduction, the authors refer to sound diffusion as the "redirection of directional sound omnidirectionally..." While this is essentially correct, it is more precise to refer to this as sound diffusion and the design of sound diffusers.
- 7) The authors refer to the transmittance law in the first paragraph and a review paper on acoustic metamaterials. What is the "transmittance law?" There is a mass law, which states that sound transmission is proportional to the product of mass and frequency, but is limited to specific frequency ranges, but there is no "transmittance law" for acoustic barriers to the knowledge of this reviewer. To that end, why do the authors refer to a 2017 reference if the law is assumed to be fundamental? Surely the law was well known prior to 2017 and a more seminal work could be referenced.
- 8) There should be a reference provided on the topic of noise cancellation headphones.
- 9) In the second paragraph, the authors should note that the first parallel between Willis media and bianisotropy was provided by Sieck et al, reference [12] of the manuscript.
- 10) The third paragraph of the introduction discusses "mechanical bianisotropy," but the manuscript and discussion is really limited to acoustic bianisotropy because it neglects any interaction with shear motion and transverse waves.
- 11) The caption of Fig 1 references a "black" particle, but there is no black particle. I believe it should be referred to as a "blue" particle.
- 12) The English of the second sentence of the Results section has grammatical errors and should be improved.
- 13) At the top of page 4 of the manuscript, the authors mention "transmissivity," which is not a term used in acoustics. It should be transmission.
- 14) The top of page 4 contains a statement that diffraction from bianisotropic materials is small, which seems to be a very broad statement that isn't necessarily supported as a general fact by the present. It definitely seems that this can be made to be true, though not necessarily in all cases, so the statement it should be qualified by the authors.

15) A recent paper by Quan et al discussing the limits of bianisotropic polarizabilities in passive materials (Phys. Rev. Lett., 120, 254301 (2018)) should be included in the references in the second sentence of the second paragraph of page 4.

16) In the same sentence, the authors state that bianisotropy takes into account "edge effects." It is not clear if this is true. Bianisotropy does include non-local effects, and therefore may account for edges, so this should be stated more clearly.

17) In the second paragraph of page 4, the authors state that passive materials require that $\alpha_{md} = \alpha_{dm}$, which is partially true. This should more specifically read passive and reciprocal materials are constrained by this.

18) The second paragraph of page 5 introduces p_b and p_c , but do not state what they represent. They are introduced in the supplementary material, which is fine, but they should be better explained in the main manuscript as well.

19) As context to the discussion on page 6 (and elsewhere, really), it would be very helpful for the authors to provide the calculation of α_m and α_d for some representative wall sections and to provide a discussion of the limits of that approximation. This would likely be best in supplementary material.

20) The transfer function "g" is discussed in some detail in the supplementary material. However, it is not clear how "g" is determined for performance at a given frequency or for a specific wall construction. It does not seem that you can simply use any impulse response "g" and get this to work, but it's possible that there is some misunderstanding here. Since this is essential to the operation of the bianisotropic surface introduced here, this must be discussed in more detail.

21) The third paragraph on page 6 discusses the physical behavior of the active bianisotropic "particle." However, this discussion does not really mention diffraction effects and how those are explicitly accounted for. Since this seems to make use of feedback (thus the discussion of system stability) more details of the control algorithm should be provided. This goes to the discussion around point 2a and active control.

22) The definition of SSL based on certain regions in space is very ad hoc and would depend on the size of the aperture in the wall, the wall aperture in general, and the frequency. Is there a more general way to define this, maybe based on some very rudimentary diffraction theory?

23) Figure 2g shows the difference in SPL calculated using Comsol. What difference is calculated? Do you run two cases in Comsol? What are they? It's possible this reviewer missed that particular detail, but I cannot find what the difference is in the manuscript.

24) The last sentence on page 7 seems to be missing the word "phase" before "gradient of p_b/p_c ."

25) Figure 3 seems to indicate that the system was designed for 3 kHz, which is related to the point raise in comment 20). Please provide some context about the design of the impulse response, g, and the target frequency.

26) The second point of the second paragraph of the Discussion section points out that the bianisotropic metasurface acts locally, which is very desirable for noise control. This point should be made more prominent in the abstract and introduction and contrast should be made with respect to existing noise control approaches, which are usually dependent on the full field or large portions of the field.

27) It seems that this does work for a barrier to eliminate transmitted acoustic waves. What about a similar type of active surface for a wall to control the interior space. The discussion should probably address this obvious question in one way or another.

28) Reference on the metamaterial diffuser is a good one, but the authors should also reference more seminal work by Schroeder on the topic.

Reviewer #2:

Remarks to the Author:

The paper proposes to achieve thin sound barriers using a Willis acoustic metamaterial slab made from active acoustic inclusions whose dipolar response is engineered using transducers controlled by an electronic circuit. I think the paper is not of very good quality, because it does not realize what it claims, and a lot of statements are simply not backed up with sufficient experimental or numerical data.

The authors go through a lot of trouble to engineer an active metasurface, and the result is that is barely does better (in terms of sound blocking) than a passive thin wall. They show that in a bandwidth (2.5 kHz to 3.5 kHz, that I would not call a broad bandwidth), the bianisotropic material does (a little bit) better than just a passive slab. However, it seems like the plot is truncated just in the right way to hide that outside of this band, the sound level goes above 0 dB, which means that the system creates some noise at other frequencies. This is a serious problem when considering that noise control is the number one motivation for this paper. In addition, the passive wall chosen is very thin and a thicker passive wall would perform much better than the bianisotropic system, and in a very broadband way.

The solution of bi-anisotropy is presented as leading to optimal efficiency. This is not justified from the theoretical side: the authors should introduce a clear metric and demonstrate that the design is optimum for this metric. Right now these theoretical claims and the experiment seem totally not connected.

Response to Reviewers

We thank the reviewers for their careful reading of our manuscript and constructive comments. We substantially revised the manuscript to address all the issues raised by the reviewers. In particular, we expanded the discussion on how the theoretical analysis on bianisotropy guides the design of the bianisotropic metasurface, and why our approach is fundamentally different from others, including traditional active sound control methods. The discussion is now supported by a new figure and new measurements of the single meta-atom behavior. In addition, we expanded the discussion on the figure of merit, the sound suppression level, which has been chosen to directly compare the sound isolation performance of the metasurface against that of a conventional wall. We also extended the bandwidth in which we present our results to the full octave 2 kHz - 4kHz. Below are our answers to all the comments raised by the reviewers.

Response to Reviewer #1

- **Comment:** *The authors content that it is solely the use of active bianisotropic elements that enable the significant improvements in sound isolations achieved in the results presented here. Indeed, equation (1) and the discussion surrounding it suggest that this is indeed that case. However, the authors have only shown the case of a finite wall with a finite window that is either of infinitely stiff and immobile “material” and therefore diffraction effects around the finite wall are present. In absence of diffraction, would this still be the case and why? The reason this question arises is that the formulation employed in Eqns. (4)-(7) imply that the wall itself acts as a scatterer of acoustic pressure. If the wall were infinite in extent and was infinitely stiff and immobile, then the impedance would be infinite and no pressure would be transmitted. This implies that the only way it is possible to improve upon the idealized finite length wall is to simply cancel the diffracted field. This seems to be the reason that the performance is somewhat narrow band and that the bandwidth of performance is dictated by the amplifier response of the active elements as indicated in Fig. 3. These concepts were not clearly fleshed out in the manuscript, though they should be since it goes to the core of the argument for the active bianisotropic metasurface. More detail needs to be provided in both the main article and supplementary material to address these points and place the behavior on more complete foundation.*

Response: Please note that we reduce the transmission coefficient through our metasurface by making it bianisotropic and *not* by making it appear very stiff and dense. Namely, we start from a conventional wall section that unavoidably allows a significant portion of low frequency sound to pass through it. The idea presented in the manuscript is to embed bianisotropic inclusions generating monopole-to-dipole polarizabilities into the wall to significantly increase the opacity of the modified wall section. To realize this idea, we designed the metasurface so that in its unpowered state (i.e. no bianisotropic response) it behaves just like the conventional wall, as demonstrated by the uniform transmission through the conventional wall with embedded unpowered metasurface (see Fig. 3e in the revised manuscript – former Fig. 2e). In its powered state the metasurface produces a bianisotropic active response that is tuned to drastically reduce the sound transmission through the metasurface.

In our approach the diffracted fields are generated exclusively by the engineered bianisotropic polarizabilities. Since we show theoretically that the metasurface bianisotropic response needs to be relatively small for perfect sound isolation, it follows that the diffraction added by the bianisotropic polarizabilities is expected and measured to be correspondingly small. Please note that we are not trying to cancel the diffracted fields propagating around an infinitely broadband stiff wall. Instead, we show that if infinitely stiff and/or massive walls existed, they would be ineffective because of significant diffracted fields generated at the wall edges. Figs. 3f and 3g in the revised manuscript show this fact by comparing the relatively uniform distribution of sound level behind the metasurface against the high amplitude variations induced by the large diffraction behind an ideal infinitely stiff and dense wall. The discussion starting with the last paragraph on page 6 explains how the theory presented on pages 3-6 guides the design and why our method is unlike other active sound isolation approaches.

Since the reviewer raised this important point, a discussion of the metasurface bandwidth was included in the revised manuscript as well (see the discussion around the newly included Fig. 2 and Fig. 4a). To summarize, the bandwidth is determined by how fast the electronics is. For convenience, we used in this proof of concept a digital circuit approach in which a significant time is wasted while the electrical signal passes through a digital-to-analog converter, an analog-to-digital converter, and a microprocessor, and even in this case the measured bandwidth is more than an order of magnitude bigger than the bandwidth of other passive and active noise mitigation systems. The electronics is very simple (amplifier followed by phase shifter) and can be made much faster. Fig.4a shows the expected broadband performance improvement if faster electronics were used.

- **Comment:** *Along the same lines, it is not clear to this reviewer that this necessarily represents an advancement over the traditional actively sound control either within a space or through apertures. Indeed, the argument that the “bianisotropic particle,” which is simply a microphone with a dipolar source, is not entirely convincing. For example, if the sensor is placed on one side of a wall of finite impedance, and then the source is embedded in the same wall, then the source acts as a monopole in each half space, obviously one that has opposite polarization on each side of the wall. It is clearly a monopole (pressure) sensor and dipolar source. However, when this is embedded in a wall structure rather than acting in free space, as is the true concept of a α_{md} polarization, then it is not clear to this reviewer that it acts as a true monopole to dipole scatterer. The argument that this configuration can be used to mimic a bianisotropic particle needs to be improved and, further, it needs to be shown that what has been shown is not simply an implementation of a standard active noise control method.*

Response: The key requirement of a bianisotropic meta-atom implementing the monopole-to-dipole polarizability is to sense the local pressure and generate a velocity field that *does not contribute to the sensed local pressure*. In other words, the meta-atom’s driver and sensor transducers need to be mechanically decoupled. i.e. there should be no direct feedback loops between the two. Our meta-atom is the first to implement this unique behavior. All active sound control methods published so far

(including the few published active metamaterial designs) employ local feedback loops between the composing sensor and driver pairs, e.g. see [M. J. Crocker, Handbook of noise and vibration control, John Wiley and Sons, 2007] for an exhaustive review of active sound and vibration control methods. The scenario proposed by the reviewer is essentially what is used in headphones with active noise cancellation and fails the no-local-feedback-loops requirement. In an active headphone, the sound produced by the on-board noise cancellation speaker couples to the sensing microphone and the resulting feedback loop needs to be controlled actively. The manuscript shows how to design the meta-atom to decouple its sensor and driver.

A second requirement of the bianisotropic meta-atom is that its acoustic response must be local, i.e. the sensor and driver are essentially collocated. This constraint assures that the metasurface is effective for various wave vector directions (for example, for the wave vectors falling in the solid angle that determines the shadow region behind the metasurface – region A in Fig. 3f). Consequently, modifying the reviewer’s scenario by placing the microphone on one side of a very thick wall and the speaker on the other can work in a uni-directional manner at best because the sound propagating through the wall between the microphone and speaker will undergo a phase advance that depends on the direction of the incident sound, while the phase advance in electronics is fixed.

Related to the second requirement, the active bianisotropic meta-atom presented in the manuscript is the first design that is very compact (its thickness is less than $\lambda/10$) and implements a local acoustic response. The few passive bianisotropic metamaterials demonstrated in the past all rely on non-local responses in larger asymmetric unit cells.

The revised manuscript now presents in the discussion surrounding Fig. 2 how the theory guides the bianisotropic meta-atom design.

- **Comment:** *This latter point is very important and also relates to the fact that the current manuscript does not address an existing literature on active noise control techniques or demonstrations. Note that it this reviewer does not believe that this manuscript would be un-publishable if the approach did have strong correlation with existing active noise control techniques since the current study investigates the potential for improving active noise control using ideas that geminated in metamaterial concepts. However, it is very important that the ultimate publication acknowledges this vast literature and is placed in the correct context in that domain.*

Response: We have now included a discussion on how our approach is related to existing active noise control techniques. As we mentioned in the previous response, the bianisotropic meta-atom must produce a velocity field that does not contribute to the local pressure field. In other words, the sensor and driver transducers are mechanically decoupled. This type of behavior has not been done before. Namely, most traditional active sound and vibration isolation solutions (see [M. J. Crocker, Handbook of noise and vibration control, John Wiley and Sons, 2007] for an excellent review of the field) have significant performance limitations imposed by their need to actively control the feedback loops between the sensing and driven transducers. They typically employ several microphones and speakers spread out in a volume. These components are

coordinated by a central computer that employs various signal processing algorithms to control in real-time the feedback loops occurring between all these elements. The complexity of these systems increase significantly with the number of microphones and speakers, thus they are not scalable in the number of speakers and microphones. Moreover, most active control systems are only designed for a given source of sound (e.g. for the narrowband sound produced by a jet engine). Consequently, traditional sound control systems are not suitable for reducing the diverse, complex, and ubiquitous noise engulfing human habitats. Moreover, the mechanical response of these systems are inherently non-local which makes performance sensitive to the incoming excitation direction. To avoid the limitations of centralized approaches, sound control methods that include decentralized, self-contained active unit cells have been proposed in the context of plate vibration control [S. G. Elliott et al, J Ac Soc Am 111, 908, 2002] and acoustic metamaterials [A. Baz, J Appl Phys 112, 084912, 2012; B.-I. Popa et al, Phys Rev B 88, 024303, 2013]. However, the feedback between the unit cell sensor and driver components is always present and reduce the effectiveness of these approaches. For instance, Elliott et al designed their feedback loops in order to avoid injection of energy into the incident wave, which reduce the vibration control efficiency significantly. This discussion has been included in the manuscript.

- **Comment:** *In regards to the structure of the bianisotropic meta-atom, it is strongly suggested that a more detailed figure showing the structure of the barrier that was tested be provided. This is very important in order to justify the claims of bianisotropy and the use of the specific configurations provided in the paper.*

Response: As the reviewer suggested, the manuscript has been revised to include the detailed description of the bianisotropic meta-atom (see Fig. 2 in the revised manuscript and the discussion surrounding Fig. 2). The metasurface is composed of five identical meta-atoms. A schematic of the metasurface that shows the position of the meta-atoms embedded into the wall is included in Fig. 3b.

- **Comment:** *The authors often refer to sound “blocking,” which seems to imply sound isolation (i.e. the elimination of transmitted sound without regard to what happens on the side of the incident sound), rather than absorption of sound. If this is the case, isolation is a preferred term and “blocking” should be eliminated from the manuscript.*

Response: We have replaced the term ”sound blocking“ by ”sound isolation“.

- **Comment:** *There are minor errors in grammar and usage throughout the text. Please do a careful read of the document prior to re-submission.*

Response: We have carefully read the manuscript and fixed numerous grammar mistakes.

- **Comment:** *The references provided in the introduction for passive sound isolation are all very new and primarily associated with metamaterials. In order to place this into the correct context, a few more seminal references on sound isolation in architectural acoustics should be provided to illustrate how long this has been a known problem.*

Response: We have followed the reviewer’s suggestions and have included more references on sound isolation. As the reviewer points out, this research area is vast. Therefore, we chose to refer to excellent presentations of the field [M. J. Crocker, Handbook of noise and vibration control, 2007; M. Long, Architectural Acoustics, Elsevier Academic Press, 2006].

- **Comment:** *In the first paragraph of the introduction, the authors refer to sound diffusion as the “redirection of directional sound omnidirectionally. . .” While this is essential correct, it is more precise to refer to this as sound diffusion and the design of sound diffusers.*

Response: We have replaced the word ”diffusers“ with ”sound diffusion“ in the first paragraph, as suggested. The phrase in which it occurs briefly explains what sound diffusion is.

- **Comment:** *The authors refer to the transmittance law in the first paragraph and a review paper on acoustic metamaterials. What is the “transmittance law?” There is a mass law, which states that sound transmission is proportional to the product of mass and frequency, but is limited to specific frequency ranges, but there is no “transmittance law” for acoustic barriers to the knowledge of this reviewer. To that end, why do the authors refer to a 2017 reference if the law is assumed to be fundamental? Surely the law was well known prior to 2017 and a more seminal work could be referenced.*

Response: The reviewer is correct, by ”transmittance law” we meant “mass law”. Refs. 2 and 3 in the revised manuscript are meant to point to past work on passive structures used as broadband sound absorbers and were not meant to refer to the mass law. The sentence was revised to make this point clear. We also included references (Refs. 2 and 3) that present the most important results of decades-long research on broadband passive materials and show how ineffective passive materials are as sound isolators.

- **Comment:** *There should be a reference provided on the topic of noise cancellation headphones.*

Response: We refer now to seminal work by Olsen and May [J Ac Soc Am 25, 1130, 1953] and B. Rafaely et al [J Ac Soc Am 102, 787, 1999].

- **Comment:** *In the second paragraph, the authors should note that the first parallel between Willis media and bianisotropy was provided by Sieck et al, reference [12] of the manuscript.*

Response: The revised manuscript acknowledges that Sieck et al draws the first parallel between Willis materials and bianisotropic media in electromagnetics.

- **Comment:** *The third paragraph of the introduction discusses “mechanical bianisotropy,” but the manuscript and discussion is really limited to acoustic bianisotropy because it neglects any interaction with shear motion and transverse waves.*

Response: We use the term ”mechanical bianisotropy“ in two places in the manuscript to refer to past work done by others on mechanical bianisotropy. Since our work

involves acoustic waves (a subset of elastic waves), we presented our work on acoustics in the more general context of elastodynamics. Moreover, our work on acoustics can be extended to control elastic waves as well.

- **Comment:** *The caption of Fig 1 references a “black” particle, but there is no black particle. I believe it should be referred to as a “blue” particle.*

Response: We thank the reviewer for pointing out this mistake. It was corrected in the revised manuscript.

- **Comment:** *The English of the second sentence of the Results section has grammatical errors and should be improved.*

Response: We have corrected the sentence in question.

- **Comment:** *At the top of page 4 of the manuscript, the authors mention “transmissivity,” which is not a term used in acoustics. It should be transmission.*

Response: We have replaced the 3 instances of the word “transmissivity” with “transmission” in the revised manuscript.

- **Comment:** *The top of page 4 contains a statement that diffraction from bianisotropic materials is small, which seems to be a very broad statement that isn’t necessarily supported as a general fact by the present. It definitely seems that this can be made to be true, though not necessarily in all cases, so the statement it should be qualified by the authors.*

Response: The reviewer is correct, not all bianisotropic materials support low diffraction. That sentence has been modified to state that the particular bianisotropic meta-surface presented in the manuscript has been designed to support low diffraction and negligible transmission coefficient.

- **Comment:** *A recent paper by Quan et al discussing the limits of bianisotropic polarizabilities in passive materials (Phys. Rev. Let., 120, 254301 (2018)) should be included in the references in the second sentence of the second paragraph of page 4.*

Response: We thank the reviewer for pointing out the reference. We have included it in the manuscript.

- **Comment:** *In the same sentence, the authors state that bianisotropy takes into account “edge effects.” It is not clear if this is true. Bianisotropy does include non-local effects, and therefore may account for edges, so this should be stated more clearly.*

Response: Please note that we have never stated that bianisotropy takes into account edge effects. Instead, the sentence to which the reviewer refers states that the microscopic view of matter that describes the response of materials to external waves in terms of polarizabilities takes into account more accurately the spatial dispersion and edge effects.

- **Comment:** *In the second paragraph of page 4, the authors state that passive materials require that $\alpha_{md} = \alpha_{dm}$, which is partially true. This should more specifically read passive and reciprocal materials are constrained by this.*

Response: We have adopted the reviewer’s suggestion.

- **Comment:** *The second paragraph of page 5 introduce p_b and p_c , but do not state what they represent. They are introduced in the supplementary material, which is fine, but they should be better explained in the main manuscript as well.*

Response: The revised manuscript explains better what p_b and p_c represent, and more importantly how these pressure fields are used to design the bianisotropic meta-atom.

- **Comment:** *As context to the discussion on page 6 (and elsewhere, really), it would be very helpful for the authors to provide the calculation of α_m and α_d for some representative wall sections and to provide a discussion of the limits of that approximation. This would likely be best in supplementary material.*

Response: The polarizabilities α_m and α_d describe the response of the conventional wall, a fairly reflective obstacle, and represent alternative measures of the bulk modulus and mass density, respectively. As previous research points out (see for example [Zigoneanu et al, J Appl. Phys. 109, 054906, 2011]) it is very difficult to accurately measure the material parameters of very thin material samples especially when these samples are mostly reflective. Moreover, the values of α_m and α_d are not very relevant for the purpose of the sound barrier (recall that our purpose is to generate the needed bianisotropic response p_b that removes the sound propagating through a conventional wall, p_c). Instead, the measure $\alpha_{md}/(1 - \alpha_m\alpha_d) \equiv p_b/p_c$ is a much more relevant quantity because an ideal sound isolator would have $\alpha_{md} = \alpha_m\alpha_d - 1$ as shown in the manuscript. In fact, $\alpha_{md}/(1 - \alpha_m\alpha_d) = p_b/p_c$ is the quantity used to design the meta-atom, and it is now plotted in the newly added Fig. 2 in the revised manuscript.

- **Comment:** *The transfer function “g” is discussed in some detail in the supplementary material. However, it is not clear how “g” is determined for performance at a given frequency or for a specific wall construction. It does not seem that you can simply use any impulse response “g” and get this to work, but it’s possible that there is some misunderstanding here. Since this is essential to the operation of the bianisotropic surface introduced here, this must be discussed in more detail.*

Response: The bianisotropic response of the meta-atom and thus metasurface is determined by the choice of electronic impulse response g . The newly introduced Fig. 2 shows that the measured pressure associated with the bianisotropic response, p_b , is proportional to the measured g in the octave of interest in which the sensing and driven transducers have approximately constant impulse responses. The meta-atom design approach presented in the discussion surrounding Fig. 2 shows how g was chosen to implement the requirement for sound isolation, namely $p_b/p_c = -1$.

- **Comment:** *The third paragraph on page 6 discusses the physical behavior of the active bianisotropic “particle.” However, this discussion does not really mention diffraction effects and how those are explicitly accounted for. Since this seems to make use of*

feedback (thus the discussion of system stability) more details of the control algorithm should be provided. This goes to the discussion around point 2a and active control.

Response: We emphasize that the active bianisotropic meta-atoms presented in this manuscript are not feedback systems, which is the major difference between our meta-atoms and other active sound control approaches reported elsewhere. To see that this is the case please recall that a bianisotropic inclusion senses the local pressure and generates a local velocity field that should not contribute to the local pressure. Furthermore, we show in the newly added Supplementary Note 3 that the indirect feedback loop caused by the driver producing sound that is scattered by the environment and couples back into the sensor is also always stable. Therefore, there is no need to implement feedback control algorithms that would limit the sound isolation efficiency as it happens to virtually all other active sound control schemes.

Moreover, we explain better in the revised manuscript why the metasurface produces lower diffraction. Namely, the idea behind the bianisotropic barrier is to embed bianisotropic inclusions in a wall region that needs to be made opaque. Since it is difficult from a practical point of view to embed the particles in the actual wall, we designed the metasurface so that it matches the the properties of the wall in its unpowered state, i.e. the metasurface has the same $\alpha_m \approx 1$ and $\alpha_d \approx 1$ as the wall. Because $\alpha_{md} = 1 - \alpha_m \alpha_d$ is relatively small, the fringe fields generated by the edges of the metasurface which form the diffracted fields are expected to be small as well. Fig. 3f in the revised manuscript confirms this expectation. This issue is clearly addressed in the revised manuscript.

- **Comment:** *The definition of SSL based on certain regions is space is very ad hoc and would depend on the size of the aperture in the wall, the wall aperture in general, and the frequency. Is there a more general way to define this, maybe based on some very rudimentary diffraction theory?*

Response: The sound pressure level (SPL) difference on which the SSL is based represents the ratio of the transmission coefficients through the metasurface and the conventional wall measured in decibels. It thus represents how much better the metasurface is at stopping sound penetration behind the barrier compared to a conventional wall. The white lines in Fig. 3f of the revised manuscript (former Fig. 2f) were obtained by uniting the speaker position with the edges of the metasurface. Therefore, region A between the two white lines (see Fig. 3f) represents the area in which we expect to see an acoustic "shadow" cast by the metasurface. The figure of merit, SSL, has been chosen to quantify how intense is this shadow cast by the metasurface by averaging the SPL difference in the shadow region behind the metasurface and comparing it with the average SPL difference measured behind the conventional wall (i.e. measured in region B). Note that regions A and B do not change with frequency and are determined entirely by the geometry of the experimental setup.

- **Comment:** *Figure 2g shows the difference in SPL calculated using Comsol. What difference is calculated? Do you run two cases in Comsol? What are they? It's possible this reviewer missed that particular detail, but I cannot find what the difference is in the manuscript.*

Response: The reviewer is correct. To obtain the SPL difference, we performed two simulations. In the first, we used a uniform wall. In the second simulation, we replaced a wall portion with an infinitely stiff and dense material. The sound pressure levels measured in these two cases were subtracted in order to obtain Fig. 2g. The manuscript was revised to clarify the simulation steps (see the Methods section).

- **Comment:** *The last sentence on page 7 seems to be missing the word “phase” before “gradient of p_b/p_c .”*

Response: We corrected this mistake in the revised version of the manuscript.

- **Comment:** *Figure 3 seems to indicate that the system was designed for 3 kHz, which is related to the point raise in comment 20). Please provide some context about the design of the impulse response, g , and the target frequency.*

Response: We have substantially revised the manuscript and we now include the design of the meta-atom and its electronics, and explain how this design was informed by the theoretical analysis.

- **Comment:** *The second point of the second paragraph of the Discussion section points out that the bianisotropic metasurface acts locally, which is very desirable for noise control. This point should be made more prominent in the abstract and introduction and contrast should be made with respect to existing noise control approaches, which are usually dependent on the full field or large portions of the field.*

Response: The reviewer raises an excellent point. We have revised the manuscript to emphasize the distinction noted by the reviewer between our approach and traditional noise control techniques in the abstract and in the main text.

- **Comment:** *It seems that this does work for a barrier to eliminate transmitted acoustic waves. What about a similar type of active surface for a wall to control the interior space. The discussion should probably address this obvious question in one way or another.*

Response: The approach shown here would work for the scenario proposed by the reviewer, but the value of α_{md} would have to approach 1, therefore the bianisotropic response would have to be much stronger and approach the level of the incident field. This is the topic of a subsequent paper.

- **Comment:** *Reference on the metamaterial diffuser is a good one, but the authors should also reference more seminal work by Schroeder on the topic.*

Response: We refer to Schroeder’s work on diffusers [J Ac Soc Am 57, 149, 1975] in the revised manuscript.

Response to Reviewer #2

- **Comment:** *The paper proposes to achieve thin sound barriers using a Willis acoustic metamaterial slab made from active acoustic inclusions whose dipolar response is engineered using transducers controlled by an electronic circuit. I think the paper is not of very good quality, because it does not realize what it claims, and a lot of statements are simply not backed up with sufficient experimental or numerical data. The authors go through a lot of trouble to engineer an active metasurface, and the result is that is barely does better (in terms of sound blocking) than a passive thin wall. They show that in a bandwidth (2.5 kHz to 3.5 kHz, that I would not call a broad bandwidth), the bianisotropic material does (a little bit) better than just a passive slab.*

Response: We disagree with both the reviewer’s statements that the bianisotropic barrier does barely better than a passive thin wall and that the barrier is not broadband. We showed experimentally that the barrier reduces the transmitted sound level by 15 dB compared to a conventional wall (see Figs. 3 and 4 in the revised manuscript). This reduction is huge. Classic textbooks on acoustics such as (Kinsler, Frey et al, Fundamentals of Acoustics, 4th edition, 2000) state the importance of the 15 dB sound level reduction. Namely, 15 dB is the difference between a quiet room and the sound produced by a vacuum cleaner (manufactured before year 2000) near its operator or the sound produced by the TV audio (See Table 13.2.2 in Kinsler, Frey et al). Moreover, M. J. Crocker (Handbook of Noise and Vibration Control, John Wiley and Sons, 2007) lists popular passive materials used to block sound in architectural acoustics (page 52, Table 2). The better passive slabs such as Rockwall of same acoustic thickness as our metasurface reduce sound by less than 3 dB in their best frequency range.

Regarding bandwidth, a sound reduction of 15 dB has currently been achieved only in very narrow bandwidths of the order of 1%-2% (see [Crocker, 2007] for an extensive review of sound and vibration mitigation techniques) in passive and active designs. In this manuscript we demonstrate a metasurface bandwidth of more than one order of magnitude larger than that. The revised manuscript discusses the factors that influence bandwidth and how the bandwidth can be enlarged beyond the experimentally measured value. To summarize, the proof of concept employs for convenience a digital approach in which a significant time is wasted while the electrical signal passes through digital-to-analog and analog-to-digital converters, and microprocessor, and even in this case the measured bandwidth is at least an order of magnitude bigger than other passive and active noise mitigation approaches. The electronics is very simple (amplifier followed by phase shifter) and can be made much faster. Fig.4a shows what to expect in terms of bandwidth increase if faster electronics were used.

- **Comment:** *However, it seems like the plot is truncated just in the right way to hide that outside of this band, the sound level goes above 0 dB, which means that the system creates some noise at other frequencies. This is a serious problem when considering that noise control is the number one motivation for this paper.*

Response: It is true that in its current implementation the metasurface slightly increases the transmission in a limited out-of-band region (2 kHz – 2.5 kHz). However

the transmission increase is small (the sound level increase is below 2 dB as seen in the revised Fig. 4 – the maximum theoretical value is only 3 dB) and can be easily removed by a simple bandpass filter placed before the driven element. The manuscript has been revised to show the the entire octave 2kHz - 4kHz.

- **Comment:** *In addition, the passive wall chosen is very thin and a thicker passive wall would perform much better than the bianisotropic system, and in a very broadband way.*

Response: There are numerous examples in which one cannot simply increase the thickness of walls to block sound. For example, building walls, doors, and windows can never be made thick enough to block low frequency sound. Car and airplane fuselages can never be made thick enough to block the road/engine noise. This is the reason why noise reduction has been a recognized problem since the 19th century, and it is still an active topic of research.

- **Comment:** *The solution of bi-anisotropy is presented as leading to optimal efficiency. This is not justified from the theoretical side: the authors should introduce a clear metric and demonstrate that the design is optimum for this metric. Right now these theoretical claims and the experiment seem totally not connected.*

Response: The manuscript introduces the sound suppression level (SSL) as the metric that measures how much better a sound barrier is compared to a conventional wall. The SSL essentially represents the ratio of the transmission coefficients through the sound barrier and the conventional wall and, ideally, it should be 0. This manuscript computes theoretically the transmission coefficients through a one meta-atom-thick metasurface and shows how bianisotropy can reduce SSL to very low values while keeping the meta-atom stable. Furthermore, the theoretical analysis shows that it is much more challenging to obtain comparable levels of SSL using non-bianisotropic materials because the active response in the latter case needs to be higher than the incident wave, which leads to instability issues. Addressing these issues is hard, as demonstrated by half a century of research on active sound and vibration control that failed to found adequate broadband solutions. The manuscript shows experimentally that the bianisotropy route is a viable solution that leads to small values of SSL unachieved in a broadband manner using other methods.

Reviewers' Comments:

Reviewer #1:

Remarks to the Author:

I appreciate the thorough response by the authors to my comments on the initially submitted manuscript. I find the revisions and responses sufficiently convincing to recommend this submission for publication as is. I do this primarily because I think the ideas that are forwarded by this work are original with respect to both classical noise control problems and novel metamaterial concepts.

Reviewer #2:

Remarks to the Author:

As the authors acknowledge, they initially swept under the rug the fact that the proposed active element actually increases the noise incident at lower frequencies. They now plot the transmission loss over a larger frequency range and reveal this crucial drawback (although they should also provide data above 4 kHz, since the curve seem to also go towards the 0dB line at higher frequencies, and may also create high frequency noise).

When I wrote my initial report, I was expecting that the authors would change their design, fix this issue and provide us with some compelling data that their noise barrier is indeed a barrier, and not a noise amplifier. I am disappointed to see that the response of the authors is just an unsupported claim that they could easily do it by adding a filter. If it is easy, why don't they do it and prove their point with new data? Actually, I think the solution of adding an extra filter is not trivial, and may have some drastic effect on the bandwidth and introduce other detrimental issues, for instance with respect to stability.

I cannot recommend for publication a work about a noise barrier that amplifies some portion of the noise spectrum. My opinion is that the paper is not acceptable for publication unless the authors provide experimental data that this issue is fixed.

Response to Reviewer #1

- **Comment:** *I appreciate the thorough response by the authors to my comments on the initially submitted manuscript. I find the revisions and responses sufficiently convincing to recommend this submission for publication as is. I do this primarily because I think the ideas that are forwarded by this work are original with respect to both classical noise control problems and novel metamaterial concepts.*

Response: We thank the reviewer for appreciating the novelty of our idea, the importance of our results, and for recommending our manuscript for publication in its current form.

Response to Reviewer #2

- **Comment:** *As the authors acknowledge, they initially swept under the rug the fact that the proposed active element actually increases the noise incident at lower frequencies. They now plot the transmission loss over a larger frequency range and reveal this crucial drawback (although they should also provide data above 4 kHz, since the curve seem to also go towards the 0 dB line at higher frequencies, and may also create high frequency noise). When I wrote my initial report, I was expecting that the authors would change their design, fix this issue and provide us with some compelling data that their noise barrier is indeed a barrier, and not a noise amplifier. I am disappointed to see that the response of the authors is just an unsupported claim that they could easily do it by adding a filter. If it is easy, why don't they do it and prove their point with new data? Actually, I think the solution of adding an extra filter is not trivial, and may have some drastic effect on the bandwidth and introduce other detrimental issues, for instance with respect to stability.*

Response: We disagree with the reviewer's suggestion that our measured 16 dB of sound attenuation does not represent "compelling data that [the] noise barrier is indeed a barrier" and that less than 2dB of sound pressure level (SPL) increase in a small region of the spectrum *outside* the band of interest makes our metasurface a "noise amplifier". We also disagree with the reviewer's suggestion that 2 dB is a significant sound level increase. A vast literature on sound perception shows that 2 dB is a very small value situated around the smallest SPL difference detected by humans [see for example page 165 of (B. C. J. Moore, An Introduction to the psychology of Hearing, 6th edition, Emerald Group Publishing, Bingley UK, 2012)]. Moreover, we demonstrate in the revised manuscript that the upper theoretical limit of the SPL increase produced by the metasurface is only 3 dB. Therefore, we believe that the effort and time needed to modify our design to remove the small measured sound level increase would not justify the improvement. As noted in our previous response to the reviewer, the modification would consist in adding one or two notch filters to the existing bandpass filter. This addition requires building a new metasurface with a new output circuit that drives the unit cell speakers. Furthermore, the reviewer is concerned that such an addition would have a detrimental effect on the stability of the metasurface. Please note that Supplementary Note 3 demonstrates that the metasurface stability does not depend on the electronics impulse response (g). Consequently, adding extra filters would have no effect on stability. The manuscript has been revised to make these points clearer.

Reviewers' Comments:

Reviewer #1:

Remarks to the Author:

My opinion has not changed regarding this manuscript and should be accepted. The paper reports works that demonstrate a novel approach to noise cancellation that is indeed limited at this stage. While it would be ideal to show improved performance, that would require additional engineering effort that would not provide much, if any, additional understanding of the fundamental physical principles of interest to the readership of Nature Communications.

Response to Reviewer #1

- **Comment:** *My opinion has not changed regarding this manuscript and should be accepted. The paper reports works that demonstrate a novel approach to noise cancellation that is indeed limited at this stage. While it would be ideal to show improved performance, that would require additional engineering effort that would not provide much, if any, additional understanding of the fundamental physical principles of interest to the readership of Nature Communications.*

Response: We thank again the reviewer for his/her constructive criticism throughout the review process, for appreciating the novelty of our idea, and for recommending our manuscript for publication in its current form.